# Increased Stiffness Downregulates Focal Adhesion Kinase Expression in Pancreatic Cancer Cells Cultured in 3D Self-Assembling Peptide Scaffolds

**DOI:** 10.3390/biomedicines10081835

**Published:** 2022-07-29

**Authors:** Nausika Betriu, Anna Andreeva, Anna Alonso, Carlos E. Semino

**Affiliations:** Tissue Engineering Research Laboratory, Department of Bioengineering, IQS-School of Engineering, Ramon Llull University, 08017 Barcelona, Spain; nausikabetriur@iqs.url.edu (N.B.); annaandreeva@iqs.url.edu (A.A.); annaalonsom@iqs.url.edu (A.A.)

**Keywords:** stiffness, biomechanics, FAK, self-assembling peptides, RAD16-I, pancreatic ductal adenocarcinoma, PDAC

## Abstract

The focal adhesion kinase (FAK) is a non-receptor tyrosine kinase that participates in integrin-mediated signal transduction and contributes to different biological processes, such as cell migration, survival, proliferation and angiogenesis. Moreover, FAK can be activated by autophosphorylation at position Y397 and trigger different signaling pathways in response to increased extracellular matrix stiffness. In addition, FAK is overexpressed and/or hyperactivated in many epithelial cancers, and its expression correlates with tumor malignancy and invasion potential. One of the characteristics of solid tumors is an over deposition of ECM components, which generates a stiff microenvironment that promotes, among other features, sustained cell proliferation and survival. Researchers are, therefore, increasingly developing cell culture models to mimic the increased stiffness associated with these kinds of tumors. In the present work, we have developed a new 3D in vitro model to study the effect of matrix stiffness in pancreatic ductal adenocarcinoma (PDAC) cells as this kind of tumor is characterized by a desmoplastic stroma and an increased stiffness compared to its normal counterpart. For that, we have used a synthetic self-assembling peptide nanofiber matrix, RAD16-I, which does not suffer a significant degradation in vitro, thus allowing to maintain the same local stiffness along culture time. We show that increased matrix stiffness in synthetic 3D RAD16-I gels, but not in collagen type I scaffolds, promotes FAK downregulation at a protein level in all the cell lines analyzed. Moreover, even though it has classically been described that stiff 3D matrices promote an increase in pFAK^Y397^/FAK proteins, we found that this ratio in soft and stiff RAD16-I gels is cell-type-dependent. This study highlights how cell response to increased matrix stiffness greatly depends on the nature of the matrix used for 3D culture.

## 1. Introduction

Pancreatic ductal adenocarcinoma (PDAC) is the most relevant pancreatic disease, accounting for more than 90% of all pancreatic malignancies. PDAC is the fourth most common cause of cancer-related death in Europe and the US, with an extremely low five-year survival rate of around 10% [1] even though its occurrence is very low (around 13 per 100,000 habitants), representing only 3% of all diagnosed cancers [1]. The high mortality of PDAC has been associated in part with its asymptomatic nature at early stages, non-specific symptoms [2] and limited response to therapy [3]. The most common driver gene mutations found in PDAC are KRAS (88–100% of cases), CDKN2A (90% of cases), TP53 (85% of cases) and SMAD4 (55% of cases), which all fulfill key roles in pancreatic tumorigenesis [4]. Apart from this genetic landscape, PDAC also has a unique tumor microenvironment (TME), which favors malignant progression as well. One of the hallmarks of PDAC, and probably the most important, is the presence of high amounts of extracellular matrix (ECM) proteins, especially type I collagen, fibronectin, laminin and hyaluronic acid, which are produced by stromal cells [5]. Interestingly, ECM production in PDAC is so large that stromal components can account for up to 90% of the tumor mass [6]. This stromal reaction causes tissue stiffening, which, in turn, also accelerates PDAC progression via changes in integrin-mediated signal mechano-transduction as well as producing a feedback loop that sustains the fibrotic activity of stromal cells. Therefore, one of the principal features of PDAC, and probably one of the main properties of the malignant phenotype, is the presence of a dense and remarkably stiff ECM surrounding the tumor, which generates a unique microenvironment that fosters sustained cancer cell proliferation and resistance to drug-induced apoptosis [7,8,9,10,11,12,13,14]. Indeed, PDAC tissue can be several folds stiffer than its normal counterpart, as determined by both ex vivo and in vivo methods [15]. For example, ex vivo studies measuring tissue stiffness by mesoscale indentation reported values of 1, 2 and 5.5 kPa (steady-state-modulus) for normal pancreas, pancreatitis and PDAC tissues, respectively [16]. In vivo studies using magnetic resonance imaging (MRE) showed that, while normal pancreas presented with an average stiffness of 2.5 kPa, tissue stiffness increased to 6 kPa for PDAC [17].

Given the significance that not only the biochemical signals but also the biomechanics have in cancer initiation and progression, researchers are progressively considering and adjusting this parameter when developing in vitro models. The classic example of an initial cancer model to incorporate matrix stiffness as a parameter was breast cancer. Since it was described 40 years ago that high breast density was associated with a four to six times higher risk of developing breast cancer, many in vitro models have been developed. Typically, these models involve the culture of cells in natural 3D matrices formed by collagen [18,19,20,21,22] or reconstituted basal membrane extract (rBM) [23,24,25], either alone or in combination with other constituents, such as alginate and gelatin. Biomechanics focused on PDAC are currently attracting the attention of more researchers [15,26], and in vitro models to assess the influence of stiffness in PDAC cells include both 2D and 3D culture systems. The 2D models include mainly cell seeding on top of polyacrylamide (PA) gels of varying stiffness coated with different ECM proteins. For example, it was reported using fibronectin-coated polyacrylamide (1–25 kPa) that increased stiffness promotes elements of epithelial-to-mesenchymal transition (EMT), such as higher vimentin expression and nuclear localization of β-catenin and YAP/TAZ in PDAC cell lines [27]. The main advantage of using coated-PA gels is that they allow the increase in substrate stiffness without increasing the number of biological signals (e.g., adhesion sites), contrary to natural matrices, such as collagen I and rBM. On the other hand, three-dimensional models include cell embedding within polymeric matrices, usually functionalized with RGD or MMP-cleavable motifs [28,29,30,31], or within ECM-like matrices, such as collagen type I and basal membrane extracts [32].

Natural matrices, such as collagen, allow the modulation of stiffness by simply changing the concentration of the gel, but they also suffer degradation in vitro due to MMPs activity [33,34], which may induce changes in fiber orientation and thickness [33]. In fact, collagen degradation by cells can start as early as 6 h in culture on polymerized collagen substrate [35]. Therefore, global measures of stiffness values in collagen gels do not necessarily represent the real local value around cells due to the potential degradation of nanofibers. Finally, the stiffness of these natural matrices is usually increased by increasing concentration, which also increases ligand density. By contrast, synthetic biomaterials promise better control of mechanical properties as they do not contain MMP-cleavable motifs per se. In particular, self-assembling peptide scaffolds (SAPS) are an attractive synthetic option for 3D cell culture. Unlike synthetic polymer scaffolds, SAPS mimic the fiber architecture found in some natural matrices, such as collagen type I gels [36]. Our group has a vast experience working with the self-assembling peptide RAD16-I (AcN-RADARADARADARADA-CONH_2_, R arginine, A alanine, D aspartic acid) [37,38]. This synthetic peptide self-assembles into a network of interweaving nanofibers of ~10 nm diameter, forming hydrogel scaffolds with 99% water content and 50–200 nm pore size, which allow to culture cells in a truly 3D environment. Importantly, the stiffness can be controlled by simply adjusting the peptide concentration, and it is likely to be maintained along culture time due to the low biodegradability of the peptide in vitro. This peptide scaffold has been demonstrated to promote adhesion [39], maintenance [40], differentiation [40,41,42,43,44] and proliferation [45,46] in a variety of mammalian cells.

In the present work, we have used RAD16-I as a 3D cell culture platform to study the effect of matrix stiffness on PDAC cells in terms of focal adhesion kinase (FAK) expression and activation by phosphorylation. We demonstrate that, even being a non-instructive milieu, RAD16-I allows integrin-mediated signal mechanotransduction, probably due to its capacity to adsorb proteins contained in fetal bovine serum (FBS) used to supplement cell culture medium. Moreover, its low biodegradability in vitro makes it an ideal platform to assess the effect of matrix stiffness per se without increasing the number of biological signals. Our results show that increased matrix stiffness in 3D cultures of RAD16-I promotes downregulation of total FAK at a protein level in all the cell types analyzed, including PDAC cell lines, breast cancer cell line MCF-7 and human normal dermal fibroblasts. Instead, the extent of FAK activation in terms of phosphorylation at position Y397 was demonstrated to be cell-type-dependent. Moreover, we found that the increased stiffness of 3D collagen gels did not affect the extent of FAK phosphorylation or promote the total FAK downregulation found in RAD16-I cultures. Altogether, our results demonstrate that the cell response to matrix stiffness depends on the cell type as well as the kind of matrix used for the 3D culture.

## 2. Materials and Methods

### 2.1. 2D Cell Culture

ANC-1 (CRL-1469, ATCC, Manassas, VA, USA), MiaPaCa-2 (85062806, ECACC, Porton Down, UK), BxPC-3 (EP-CL-0041, Elabscience, Houston, TX, USA), MCF-7 (EP-CL-0149, Elabscience) and hNDF (C-12302, Promocell, Heidelberg, Germany) were cultured at 10,000 cells/cm^2^ for no more than 15 passages in DMEM (DMEM-HXA, Capricorn, Ebsdorfergrund, Germany) or RPMI (RPMI-XA, Capricorn) (in the case of BxPC-3 cells) supplemented with 10% FBS (S1810, Biowest, Nuaillé, France), L-glutamine (X055, Biowest) and Penicillin/Streptomycin (L0022, Biowest). Cultures were maintained at 37 °C and 5% CO_2_ in a humidified atmosphere.

### 2.2. 3D Cell Culture in the Self-Assembling Peptide Scaffold RAD16-I

Three-dimensional (3D) cell cultures were prepared using the synthetic self-assembling peptide scaffold RAD16-I [38], commercially available as PuraMatrix^TM^ (354250, Corning, New York, NY, USA). The peptide stock (1% in water) was diluted to a final concentration of 0.3% (*v*/*v*) in 10% (*w*/*v*) sucrose (S0389, Merck, St. Louis, MO, USA) or maintained at 1% (stock) and sonicated for 30 min. Meanwhile, cells were harvested by trypsinization and resuspended to 4 × 10^6^ cells/mL in 10% (*w*/*v*) sucrose, which is an isotonic and non-ionic medium that avoids peptide spontaneous assembly during the encapsulation process. The cell suspension was then mixed with an equal volume of 0.3% or 1% RAD16-I peptide solution to obtain a mixture of 2 × 10^6^ cells/mL and 0.15% (soft) or 0.5% (stiff) RAD16-I. Next, 40 µL of cell/peptide suspension (80,000 cells) was loaded into wells of a 48-well plate previously filled with 500 µL of culture medium, which induced the peptide spontaneous self-assembly. The plate was left in the flow cabinet for 20 min to let the peptide gel and then placed in the incubator for 1 h. Medium was changed twice to favor the leaching of sucrose. The 3D cultures were maintained in DMEM supplemented with 10% FBS, L-glutamine and P/S at 37 °C and 5% CO_2_ in a humidified atmosphere, and medium was changed three times per week. Cells were cultured for 6 days to ensure adaptation to the 3D environment. For RAD16-I functionalization, 90 µL of cell suspension was mixed with collagen type I (stock solution 3 mg/mL, A1048301, Gibco) in 97:3, 95:5 and 90:10 proportions, and then mixed with an equal volume of RAD16-I peptide solution and induced to self-assemble by contact with culture medium.

### 2.3. 3D Cell Culture in Collagen Type I

Three-dimensional cultures (3D) cultures were prepared using high concentration rat tail collagen I (10 mg/mL, 354249, Corning) at a final concentration of 1.5 mg/mL or 5 mg/mL. Collagen I was mixed with Phenol red and PBS and brought to alkalinity using NaOH, maintaining the solution in ice to avoid collagen gelation. Cells were harvested by trypsinization and resuspended to 5 × 10^6^ cells/mL in 1x PBS. Then, the cell suspension was mixed with the collagen solution, and 40 μL of the mixture (80,000 cells) was loaded into 24-well plates. Collagen constructs were left gelling for 40 min in the incubator (37 °C). Medium was then added to the wells, and cultures were placed in the incubator overnight. Next day, 3D constructs attached to the bottom of the well were released with a cell lifter. Medium was changed three times per week. Cells in collagen were cultured for 6 days to ensure adaptation to the 3D environment. Further, 10 µM GM6001 (sc-203979, scbt, Dallas, TX, USA) was added to culture medium 24 h after cell embedding in collagen to inhibit MMP activity.

### 2.4. Protein Binding to RAD16-I Gels

To test protein binding to cell-free RAD16-I gels, 100 µL of a 0.3% peptide solution was loaded into tissue culture inserts, gelled for 1 h with 10% FBS in PBS or PBS alone (negative control), and then washed with PBS overnight several times. Gels were disrupted in RIPA buffer, and bound protein content was quantified with a BCA assay (39228, Serva). To test collagen binding to RAD16-I gels, 10 μL of a 3 mg/mL collagen solution (A1048301, Gibco) was mixed with 90 μL of 10% sucrose and afterwards mixed with an equal volume of a 0.6% RAD16-I peptide solution. Further, 40 μL of this solution was induced to self-assemble with PBS for 1 h, and then washed with PBS overnight several times. Gels were disrupted by vortexing in 2x SDS sample loading buffer (LC2676, Thermo Fisher, Waltham, MA, USA), mixed with β-mercaptoethanol at 10% final concentration and boiled for 5 min at 95 °C. Samples (20 μL) were loaded in 10% Tris-Glycine pre-cast protein gels (XP00105BOX, Thermo Fisher) and run by applying 120 V for 90 min. Finally, gels were stained with Coomassie blue and distained overnight.

### 2.5. MTT Assay for Cell Viability and Proliferation

MTT [3-(4,5-dimethylthiazol-2-yl)-2,5-diphenyltetrazolium bromide] (M5655, Merck) was used to assess cell viability in 3D cultures. Cell culture medium was aspirated and 500 μL of MTT reagent was added to a final concentration of 0.5 mg/mL in culture medium. Samples were incubated for 3 h at 37 °C and 5% CO_2_ in a humidified atmosphere. MTT solution was then removed, and cells were lysed with 200 µL of DMSO (D8418, Merck). Absorbance was read at 570 nm using a microplate reader (Biotek Epoch^TM^, Biotek, Winooski, VT, USA).

### 2.6. Immunofluorescence and Image Analysis

Cells in 2D and 3D cultures were fixed with 3.7% formaldehyde containing tyrosine phosphatase inhibitor (ab201113, abcam, Cambridge, UK) at 1:100 for 15 min and washed with 1× PBS. Cultures were blocked with 5% BSA/0.1% Triton X-100 in PBS for 1 h (2D cultures) or 2 h (3D cultures) and incubated overnight at 4 °C with the following primary antibodies: anti-β1 integrin (ab24693, abcam) at 1:500, anti-pFAK^Y397^ (700255, Invitrogen, Waltham, MA, USA) at 2:500, anti-pFAK^Y861^ (44-626G, Invitrogen) at 2:500, anti-vinculin (700062, Invitrogen) at 2:500 and anti-fibronectin (14-9869-80, Invitrogen) at 1:500 in 1% BSA. Next, cells were extensively washed with 1% BSA and incubated for 2 h with secondary antibodies conjugated to A488 (ab150105, abcam) and A647 (ab150079, abcam) at 1:500. Finally, samples were counterstained with Phalloidin-TRITC and DAPI for cytoskeleton and nuclei visualization. Pictures were acquired with Leica Thunder Imager widefield microscope (Leica Microsystems, Wetzlar, Germany) coupled to a Leica DFC9000 GTC sCMOS camera using an APO 63× objective. Images were processed and analyzed with ImageJ software version 2017-05-30 (NIH, Bethesda, MD, USA) [47].

For colocalization analysis, Mander’s colocalization coefficients (MCCs) were calculated. MCC is an overlapping parameter that describes the proportion of channel A signal coinciding with channel B over the total A intensity (M_1_) and the proportion of channel B signal coinciding with channel A over the total B intensity (M_2_) [48]. This coefficient ranges from 0 (no overlapping) to 1 (total overlapping). The analysis was performed using ImageJ software [47]. Each channel was processed for background subtraction and filtered using Median and Gaussian Blur to reduce the presence of noise. Images were then segmented by thresholding using the Default option, and the resulting binary images were cleaned with the Erode and Open functions. Binary images were then used as masks to sample the denoised images using the Image Calculator function with the Min operator, creating a background-less image for each channel [49]. Finally, Mander’s colocalization coefficients were calculated using JACoP (Just Another Colocalization Plugin) version 2.0 [50], setting the threshold values to 1. Coefficients were obtained from at least 5 images.

### 2.7. Western Blot

Cells in 2D cultures and 3D constructs were lysed with RIPA buffer (R0278, Merck) containing protease inhibitor cocktail (11836153001, Roche, Basel, Switzerland) and phosphatases inhibitor (ab201113, abcam) at 1:100. Total protein content was quantified with a BCA protein assay kit (39228, Serva, Heidelberg, Germany), and 5 μg of protein was loaded into polyacrylamide gels and run by applying 120 V for 90 min. Afterwards, proteins were transferred to a PVDF membrane (IPVH07850, Merck) by applying 40 V for 2 h. The membrane was then blocked for 1 h with 4% (*w*/*v*) nonfat powdered milk in 0.2% PBS-Tween (for non-phosphorylated proteins) or 5% BSA in 0.1% TBS-Tween (for phosphorylated proteins). Next, the membrane was incubated overnight at 4 ºC with the following primary antibodies: anti-pFAK^Y397^ (700255, Invitrogen) at 1:1000, anti-FAK (AHO0502, Invitrogen) at 1:500, anti-vinculin (700062, Invitrogen) at 1:1000, anti-pErk1/2^T202,Y204^ (14-9109-80, Invitrogen) at 1:500, anti-Erk (686902, Biolegend, San Diego, CA, USA) at 1:1000 and anti-GAPDH (649201, Biolegend) at 1:2000 in 4% milk for non-phosphorylated or 1% BSA for phosphorylated proteins. The membrane was then washed and incubated with secondary antibodies anti-rabbit-HRP (ab6721, abcam), anti-mouse-HRP (ab6820, abcam) and anti-rat-HRP (405405, Biolegend), all of them at 1:1000 for 1 h at RT. Finally, the membrane was revealed for HRP detection with a SuperSignal West Pico Chemiluminescent Substrate (34080, Thermo Scientific). Chemiluminescent images were taken in the ImageQuant^TM^ LAS 4000 mini (GE HealthCare, Chicago, IL, USA). Protein bands were quantified using ImageJ software and expressed as a ratio between the protein of interest and the loading control. Each blot was repeated three times (N = 3).

### 2.8. Statistics

Data are presented as mean ± standard deviation. Conditions were tested in triplicate (n = 3) in three independent experiments (N = 3). Statistical differences were analyzed with GraphPad Prism 6 (San Diego, CA, USA) by one-way or two-way ANOVA, followed by Tukey’s multiple comparisons test. Statistical differences were indicated as * for *p* value < 0.05, ** for *p* value < 0.01, *** for *p* value < 0.001 and **** for *p* value < 0.0001.

## 3. Results

### 3.1. Cell Culture in 3D Self-Assembling Peptide Scaffold RAD16-I

Three-dimensional cell cultures were prepared using the self-assembling peptide scaffold RAD16-I as a synthetic matrix for cell embedding. We established both soft and stiff 3D environments by changing the peptide final concentration: peptide hydrogels at 0.15% would have a storage modulus (G’) of approximately 100 Pa, while hydrogels at 0.5% have a stiffness of ≈2500 Pa, previously measured by rheometry (Appendix A) [36]. Cells in 2D cultures were harvested and mixed with the peptide solution, which self-assembles into a nanofiber network upon contact with the cell culture medium, resulting in cells embedding in a truly 3D environment (Figure 1a) in macroscopic size constructs of around 5 mm diameter and 0.5 mm thickness (Figure 1b). Even though RAD16-I is a non-instructive matrix from the point of view of receptor recognition/activation, cells under both 3D environments (0.15% and 0.5% RAD16-I) were able to spread and interact with each other and with their surrounding matrix (Figure 1a). Moreover, tumor cells cultured in 3D were able to proliferate at the same rate regardless of matrix stiffness (Figure 1c).

This self-assembling peptide is also protein-adsorbent [39]. Although it does not contain integrin-binding sites, and cannot, therefore, mediate ligand-dependent ECM receptor signaling per se, it can adsorb ECM proteins, such as laminin [51], reconstituted basal membrane (rBM) [51] and fibronectin [39]. Moreover, the proteins contained in the fetal bovine serum (FBS) (most probably vitronectin and fibronectin [52]) used to supplement cell culture medium may also be responsible for cell adhesion to the peptide nanofibers [39]. We also hypothesized that cells in RAD16-I matrices could decorate their own environment by synthesizing their own extracellular matrix proteins. To confirm that proteins contained in the serum remained attached to the hydrogel, we assessed the protein content in cell-free RAD16-I gels incubated with 10% FBS (Figure 2a). We also performed fibronectin immunostaining of pancreatic cancer cells cultured in RAD16-I matrices (Figure 2b). We found that PANC-1 and MiaPaCa-2 cells in 3D RAD16-I hydrogels were also able to deposit and decorate their cellular environment with fibronectin (Figure 2b). Moreover, we detected the presence of fibronectin in cell-free regions of the 3D matrix, further confirming that fibronectin in the FBS was adsorbed into the hydrogel.

### 3.2. RAD16-I Scaffold Allows FAK Activation Independent of Matrix Stiffness

At a cellular level, matrix recognition and stiffness are sensed through integrins, which, after engagement with the ECM, activate different downstream signaling pathways, such as the focal adhesion kinase (FAK) and the extracellular-signal-regulated kinase/mitogen-activated protein kinase (ERK/MAPK) [53]. Moreover, proteins involved in signal mechanotransduction, such as β1-integrin and its downstream effector, FAK, have been known to be more active in stiff matrices compared to soft ones in a variety of cancer cells, such as gastric cancer cells and hepatocellular carcinoma cells [54,55]. Cells cultured in 2D plastic dishes, which are extremely stiff (G’ ≈ 2.5 GPa), have been described to recruit vinculin to integrins and phosphorylate FAK at position Y397 as well as to present with actin stress fibers [25]. In accordance with previous studies [25], we found that cells in classic 2D monolayer culture on plastic dishes phosphorylated FAK at positions Y397 and Y861 (Appendix A). Moreover, β1-integrin colocalized with vinculin and actin, confirming that, under stiff 2D culture conditions, integrins are active and engaged with ECM proteins (Appendix A).

To confirm that, even being a non-instructive milieu, RAD16-I was able to trigger integrin-mediated signal transduction, we performed immunofluorescence and co-localization analysis of β1-integrin and FAK in soft (0.15% RAD) and stiff (0.5% RAD) 3D matrices. We hypothesized that the fibronectin adsorbed into the hydrogel (containing RGD motif) would be enough to activate integrins and promote FAK activation. Indeed, we could detect the active form of FAK (pFAK^Y397^) in 3D cultures in both soft and stiff matrices as well as pFAK^Y861^ (Figure 3). Moreover, in both 3D cultures, vinculin was detected co-localizing with β1-integrin and actin in both stiffness conditions, similar to 2D cultures. Interestingly, in PANC-1 cells, the colocalization degree of pFAK^Y861^ with β1-integrin was higher in 3D stiff and 2D cultures than 3D soft cultures, while pFAK^Y397^ in soft 3D cultures co-localized more with β1-integrin than in stiff conditions. On the other hand, no statistical differences were found in MiaPaCa-2 cells between 3D and 2D cultures.

### 3.3. Increased Stiffness Promotes FAK Downregulation in RAD16-I Scaffold

We next analyzed the expression of FAK and its active form (pFAK^Y397^) in 2D cultures and 3D soft and stiff RAD16-I matrices by Western blot (Figure 4). We performed these experiments with the PDAC cell lines PANC-1 (epithelial/mesenchymal phenotype) and MiaPaCa-2 (mesenchymal phenotype) [32] as well as non-tumoral human dermal fibroblasts (hNDF, mesenchymal phenotype). The results show that 2D cultures, which are extremely stiff and out of the pathophysiological range (G’ ≈ 2.5 GPa), promoted the highest total and active FAK expression. Surprisingly, stiff RAD16-I 3D cultures (≈2500 Pa) promoted total FAK downregulation in the three cell types analyzed (PANC-1, MiaPaCa-2 and hNDF) (Figure 4a,b). However, the phosphorylation pattern of FAK was different for the two PDAC cell lines. In particular, PANC-1 cells in soft 3D matrices (100 Pa) had higher levels of both total and phosphorylated FAK, and, therefore, when representing the ratio pFAK^Y397^/FAK, we found approximately a ≈two-fold downregulation in stiff matrices compared to soft ones (Figure 4b, black bars). This pattern was reversed in MiaPaCa-2 cells. In this case, the cells showed similar levels of pFAK^Y397^ in both soft and stiff 3D matrices, and higher levels of total FAK in soft matrices. Therefore, when representing the ratio pFAK^Y397^/FAK, we found that, proportionally, MiaPaCa-2 in stiff matrices phosphorylated to a higher extent the FAK (Figure 4b, gray bars). Fibroblasts in 2D cultures and soft 3D cultures showed a similar pattern as PANC-1 cells, but, in stiff 3D cultures, both FAK and its phosphorylated form were almost undetectable (Figure 4a). Increased stiffness in RAD16-I 3D cultures also promoted total FAK downregulation in another PDAC cell line (BxPC-3) and in another well-known epithelial breast cancer cell line, MCF-7 (Appendix A). In view of the unexpected results obtained when working with the two stiffnesses of ≈100 Pa (0.15% RAD) and ≈2500 Pa (0.5% RAD), we also analyzed an intermediate (0.3% RAD, 700 Pa) and a higher stiffness (0.8% RAD, 8500 Pa) in PANC-1 cells (Appendix A). Under these conditions, Western blot bands further confirmed that total FAK expression and its activated form (pFAK^Y397^) proportionally decreased with increased matrix stiffness (Figure 4c,d).

To determine the effect of biological signaling on FAK expression, we also functionalized RAD16-I matrices with small amounts of collagen type I, named RAD/COL gels. Collagen was successfully incorporated into RAD16-I matrices by simply mixing and has retained after several overnight washes, as demonstrated by SDS-PAGE of cell-free gels (Figure 5a), which revealed the typical pattern for type I collagen with bands at approximately 215 kDa (β-chain), 130 kDa (α1 chain) and 115 kDa (α2 chain). We also detected an intense band in the bottom of the gel, corresponding to the RAD16-I peptide, with a molecular weight of approximately 2 kDa [37]. Moreover, fibroblasts in RAD/COL gels could detect the presence of collagen, therefore spreading and interconnecting to each other faster than those in RAD cultures (Figure 5b).

Western blot analysis of PANC-1 cells in functionalized matrices showed that the addition of collagen into RAD16-I gels did not induce significant differences in protein expression or reverse the soft/stiff pattern for FAK and pFAK^Y397^ found in the non-functionalized RAD16-I matrix (Figure 5c,d). Moreover, Erk1/2 was found active in both stiffness conditions and no difference in their expression was found between non-instructive and collagen-functionalized matrices (Figure 5c,d). The same pattern was maintained when co-culturing PANC-1 cells with dermal fibroblasts in RAD16-I 3D matrices (Appendix A). Therefore, we hypothesized that the effect that the addition of small amounts of collagen could have on FAK expression was probably overlapped by the proteins present in the FBS or the ones produced by its own cells (Figure 2).

### 3.4. FAK Expression and Activation Is Maintained Constant Regardless of Stiffness in 3D Collagen Type I Cultures

We next wanted to determine whether FAK downregulation found in 0.5% RAD16-I matrices was also observed in a classic gold standard 3D culture system, such as collagen type I gels. For that, we cultured the same cells in 3D collagen gels with concentrations of 0.15% and 0.5%, which have an associated stiffness of approximately 100 Pa and 3000 Pa, respectively, determined by dynamic mechanical analysis (DMA) [56]. Spontaneous contraction of the 3D matrix was detected in soft (0.15%) collagen gels due to the strong interaction of cells with collagen fibrils through integrins [57], most probably α2β1 [57], which enabled the cells to drag the hydrogel (Figure 6a). This phenomenon was not observed in stiff (0.5%) collagen gels, probably because the cells could not overcome the surrounding resistance to drag the hydrogel with them, which proved the dependence between matrix contraction and stiffness. Surprisingly, we found that, in this case, PDAC cells did not respond to the increased matrix stiffness as no differences in total or active FAK expression were found (Figure 6b,c). This phenomenon was also reproduced in MCF-7 breast cancer cell line (Figure 6b,c).

Even though matrix stiffness was maintained along culture time (10 days) (Appendix A), global measures of stiffness values did not necessarily represent the real local value around cells due to the potential degradation of nanofibers by cells. We hypothesized that, if cells would not be able to degrade the collagen fibers, the downregulation pattern previously observed for RAD16-I would be reproduced in collagen gels. To address this, we incubated the cells with the MMP inhibitor GM6001, but it did not cause any effect on the pattern of FAK expression or phosphorylation (Figure 6d), suggesting that potential collagen degradation by MMP was not affecting FAK expression or phosphorylation.

## 4. Discussion

A common feature in some cancers is an increased deposition of ECM components, mainly collagen I, which is in part responsible for the increased stiffness in these tumors. Indeed, tumor tissues, such as breast cancer [25], colorectal cancer [58], pancreatic ductal adenocarcinoma (PDAC) [27] and hepatocellular carcinoma (HCC) [59], can be several folds stiffer than their healthy counterparts. Matrix stiffness influences cell fate and behavior, and, therefore, researchers are progressively adjusting this parameter when performing experiments. Cancer models that included stiffness as a parameter have demonstrated that it can regulate epithelial-to-mesenchymal transition (EMT) [27,60,61,62,63], resistance to chemotherapy [19,31,55], cell proliferation [18,30,31,55,62], migration [24,64,65] and invasion [19,20,21,22]. However, the use of different matrices or the experimental temporality can yield contradicting results. For example, increased stiffness has shown to promote proliferation in natural scaffolds, such as collagen type I [18,54] and alginate/Matrigel [62], while restricting proliferation in cells cultured in polymeric scaffolds [30,31]. Studies using synthetic matrices in a physiological stiffness range (0.1–20 kPa) reported that invasion is mostly promoted with increased stiffness, but it is inhibited at nonphysiological higher stiffness (>20 kPa) [66]. Moreover, matrix stiffness has been shown to delay cell invasion [22], and, therefore, studies with short endpoints [20] may not reflect the actual infiltration of cancer cells at later points, and, therefore, these studies likely do not report that stiffness promotes invasion [66]. Therefore, the results should be interpreted in the context of each experimental design, considering not only the stiffness range but also other parameters, such as the type of three-dimensional matrix used and experimental temporality [66].

Focal adhesion kinase (FAK) is a nonreceptor protein tyrosine kinase that plays a key role in tumor invasion and metastasis. FAK is a key component present in focal adhesions [67], and it is known to be sensitive to stiffness, becoming activated when recruited to focal adhesions after integrin engagement with the ECM in stiff environments. FAK is hyperactivated in several cancers, including PDAC, and it is correlated with high levels of fibrosis [68]. In vitro models to study the effect of matrix stiffness in terms of FAK activation have been developed mainly in the context of breast cancer as mammographically dense breast tissue is one of the greatest risk factors for developing breast carcinoma. In the present work, our aim was to develop a 3D in vitro model to study the effect of matrix stiffness in PDAC cells mainly in terms of FAK expression as such models do not exist up to date. For that, we used a synthetic nanofiber matrix, RAD16-I, which, importantly, does not suffer a significative degradation in vitro in contrast to matrices from natural origin, such as collagen type I [35]. For example, pancreatic cancer cell lines cultured in 3D collagen gels upregulate MT1-MMP expression, which produces changes in collagen fiber orientation and thickness as a consequence of local collagen degradation around the cells [33], therefore compromising the maintenance of the same mechanical properties along culture time. Synthetic hydrogels, such as the ones formed by self-assembling peptides, allow one to maintain the same experimental conditions in terms of matrix stiffness, which is vital to study signal mechanotransduction pathways in a 3D context.

The RAD16-I matrix does not contain biological motifs, and, therefore, it is non-instructive from the point of view of receptor recognition/activation. However, RAD16-I has been proven to be protein-adsorbent (Figure 2a). In particular, we detected the presence of fibronectin, which could be both soluble fibronectin from the FBS used for medium supplementation and/or fibronectin secreted by the cells (Figure 2b). Fibronectin, as well as other ECM proteins, is usually secreted by resident cells in stromal compartments, such as pancreatic stellate cells (PSCs) [14]. However, pancreatic epithelial cells have also been described to synthesize and secrete fibronectin as part of their malignant transformation during tumor initiation and progression [51]. Consistent with previous results reporting fibronectin synthesis in pancreatic cancer cells cultured in 3D collagen and Matrigel hydrogels [32,33], PDAC cells in RAD16-I were also able to decorate their environment with fibronectin (Figure 2). Therefore, it is likely that fibronectin or other RGD-containing proteins present in the FBS or synthesized by the cells and adsorbed into the RAD16-I hydrogel are mediating integrin signaling in such a 3D system. In fact, our results show that vinculin and actin are recruited to β1 integrin adhesions in both soft and stiff 3D environments, similar to 2D cultures, and also that FAK activation occurs in both stiffness conditions (Figure 3). This contrasts with previous studies that show that only mammary epithelial cells (MECs) within a stiff matrix can phosphorylate FAK^Y397^ and recruit vinculin to β1 integrin adhesions [25]. Interestingly, the same study reports that MECs in soft matrix express higher amounts of total and active Src family kinases (Src, Lyn and Lck) [25] compared to stiff matrices, while others propose the contrary [18]. In our case, cells in a soft synthetic matrix expressed higher amounts of total FAK than in stiff matrices in all cell types analyzed, while phosphorylation at position Y397 and the resulting pFAK^Y397^/FAK ratio was cell-type-dependent (Figure 4).

We next wanted to test if the binding of distinct specific integrins to different ECM proteins could be the reason for these results. For that, we functionalized RAD16-I with small amounts of collagen type I. Collagen has previously been incorporated into Fmoc-based self-assembling peptide hydrogels by simple diffusion [69] and has been shown to interact with the hydrogel fibers without affecting the overall mechanical properties. Moreover, collagen molecules incorporated into the hydrogels were biologically active and provided sites for adhesion through interaction with the α2β1 integrin [69]. We have successfully incorporated collagen type I into RAD16-I hydrogels by simply mixing the collagen solution with the peptide before inducing the self-assembling. Collagen was retained in the hydrogel after several washes, as demonstrated by SDS-PAGE (Figure 5a). Moreover, fibroblasts spreading in RAD/COL gels demonstrate that collagen was still biologically active and provided sites for cell adhesion (Figure 5b). However, the addition of collagen into RAD16-I hydrogels had no effect on FAK expression as the same downregulation pattern with increased stiffness obtained for RAD16-I hydrogels was observed (Figure 5c,d).

To address these unexpected results, we cultured the cells in classic 3D full collagen type I hydrogels. Surprisingly, we found that neither PDAC cells nor breast cancer cells responded to the increased matrix stiffness as the total and active FAK levels were maintained constant between both conditions (Figure 6). Importantly, while PANC-1 cells express the collagen-recognizing integrin α2β1, MiaPaCa-2 cells lack both integrins that bind to collagen (α1β1 and α2β1) and did not attach or proliferate when cultured on top of type I collagen gels [70]. However, both cell lines were able to phosphorylate FAK at Y397 independent of their integrin expression pattern. Therefore, it is probable that serum proteins, such as fibronectin, are also adsorbed into collagen fibers [71,72], and cells are actually recognizing and binding through RGD-containing proteins and not directly to collagen fibers.

It has recently been suggested that more confining matrix architectures reduce cell adhesions to the matrix and decrease pFAK^Y397^ levels [73]. Increasing peptide concentration significantly affects the stiffness of the resulting self-assembling peptide gel; however, the structure of the nanofibers, such as pore size, may also be changing [36,51], and, therefore, cells in stiffer hydrogels are under more constraining conditions. While cells in such synthetic environments cannot degrade the nanofibers due to the absence of degradation sites, cells within 3D collagen gels can rely on enzymatic proteolysis to remodel their surrounding matrix and relieve constraints [33]. We hypothesized that, because cells in collagen matrices could be locally degrading the collagen fibers, cells in stiffer matrices may sense the same stiffness that exists in softer matrices as a result of matrix degradation, and, for this reason, no changes were detected at the level of FAK phosphorylation or expression. To address this, we incubated PANC-1 cells with a broad-spectrum MMP inhibitor (Figure 6d), expecting an increase in pFAK^Y397^ expression in stiff collagen gels, but no differences were detected. Therefore, collagen degradation by cell-secreted MMPs was probably not the reason that FAK downregulation found in stiff RAD16-I cultures was not being reproduced in stiff collagen gels. Other hypotheses that could explain these differences between synthetic and natural 3D matrices could be integrin co-signaling competition events between fibronectin and collagen as well as differences in hydrogel nanostructure features between collagen type I nanofibers and self-assembling peptide scaffold networks. In fact, it has been reported that RAD16-I gels self-assembled in longer nanofibers and formed an increased amount of cross-links and fibril entanglements compared to collagen type I [74].

## 5. Conclusions

We have developed a 3D cell culture model using a synthetic peptide-based matrix to study the effect of matrix stiffness on PDAC and other cells in terms of FAK expression. Even though self-assembling peptide scaffolds closely mimic the fiber architecture found in some natural matrices, such as collagen type I gels, the cell response to increased stiffness strongly differed in both types of matrices. This study highlights that the cell response to increased stiffness strongly depends on the type of matrix used for the 3D culture.

## Figures and Tables

**Figure 1 biomedicines-10-01835-f001:**
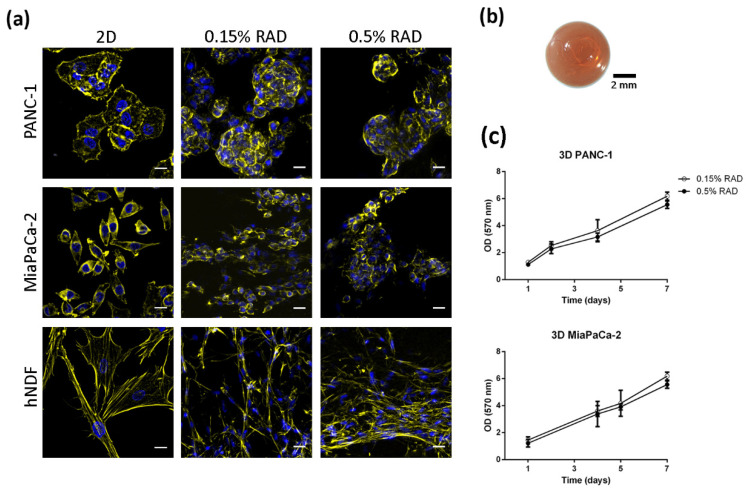
Cell culture in 3D RAD16-I scaffolds. (**a**) Cell morphology in 2D and 3D cultures. Actin (Phalloidin, pseudo-colored in yellow) and nuclei (DAPI, blue) staining of PANC-1, MiaPaCa-2 and hNDF cells cultured in classic 2D dishes and RAD16-I 3D scaffolds at two different peptide concentrations. Scale bars represent 20 µm; (**b**) macroscopic view of 3D constructs stained with Congo red for clearer visualization; (**c**) growth curves of PANC-1 and MiaPaCa-2 cells in 0.15% and 0.5% RAD16-I.

**Figure 2 biomedicines-10-01835-f002:**
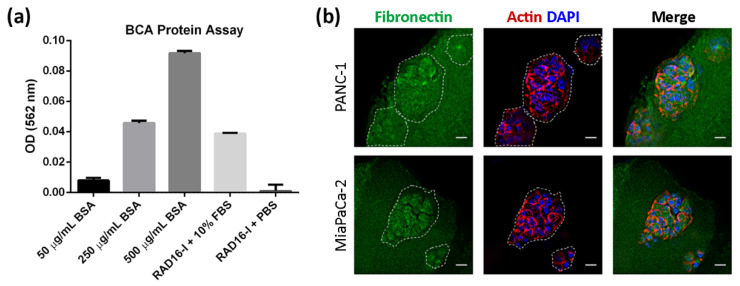
Protein adsorbing into RAD16-I hydrogels. (**a**) BCA protein assay of RAD16-I gels. The graph shows the OD values obtained for known concentrations of protein (50 μg/mL−500 μg/mL BSA) and RAD16-I hydrogels assembled with 10% FBS or PBS; (**b**) fibronectin (green) immunofluorescence of PANC-1 and MiaPaCa-2 cells in RAD16-I 3D culture counterstained with phalloidin (red) and DAPI (blue). Scale bars represent 20 µm.

**Figure 3 biomedicines-10-01835-f003:**
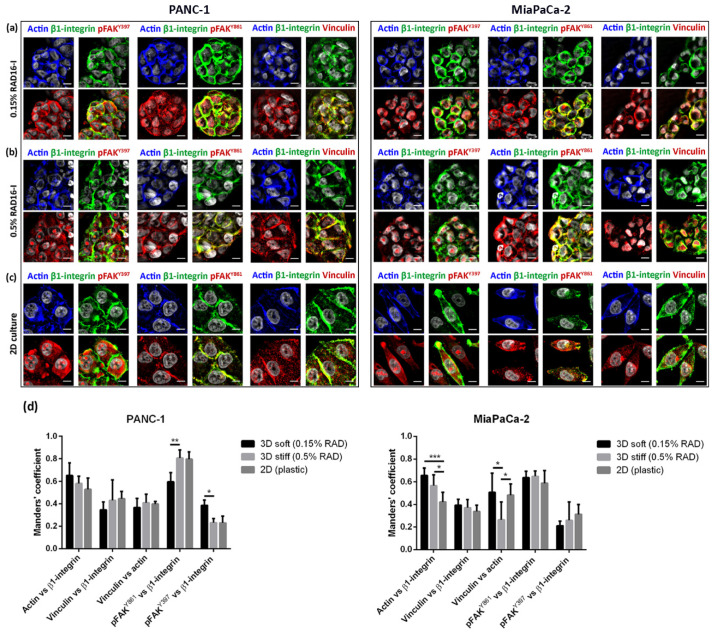
Immunofluorescence analysis of signal mechanotransduction proteins. Actin (blue) β1-integrin (green), pFAK^Y397^ (red), pFAK^Y861^ (red) and vinculin (red) staining in PANC-1 (left) and MiaPaCa-2 (right) cells in (**a**) 0.15% RAD16-I; (**b**) 0.5% RAD16-I and (**c**) 2D cultures. Merge of green and red channels is also shown. Scale bars represent 10 µm; (**d**) Mander’s colocalization coefficients of PANC-1 and MiaPaCa-2 cells (n = 5). Statistical differences are indicated as * for *p* value < 0.05, ** for *p* value < 0.01 and *** for *p* value < 0.001.

**Figure 4 biomedicines-10-01835-f004:**
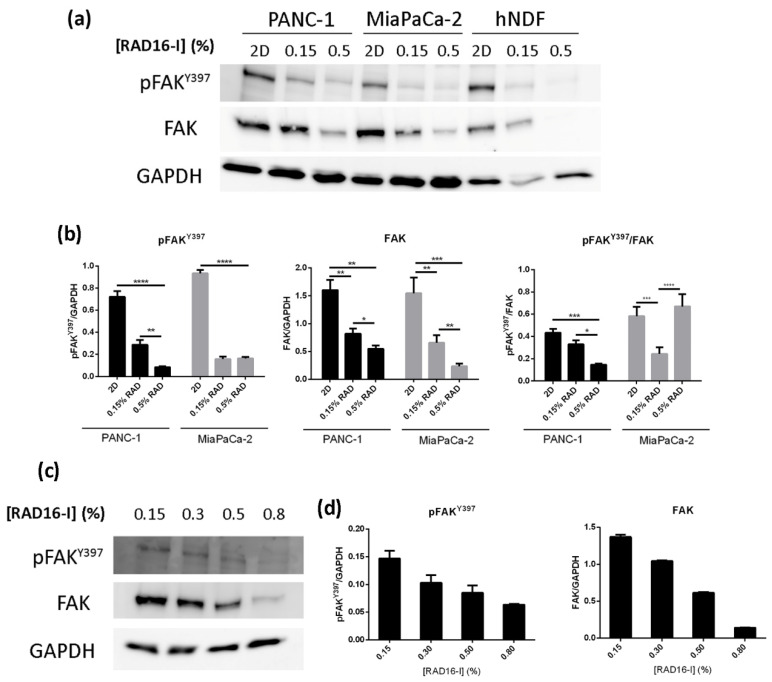
pFAK^Y397^ and FAK expression in cancer cell lines and normal fibroblasts cultured in RAD16-I hydrogels. (**a**) Western blot bands of pFAK^Y397^ and total FAK in PANC-1, MiaPaCa-2 and hNDF cells cultured in 2D and 0.15% (soft) and 0.5% (stiff) RAD16-I gels; (**b**) densitometry of bands shown in (**a**); (**c**) Western blot bands of pFAK^Y397^ and total FAK in PANC-1 cells cultured in 3D RAD16-I hydrogels of increased stiffness (from 0.15% to 0.8% RAD16-I, representing a stiffness range of 100–8500 Pa); (**d**) densitometry of bands shown in (**c**). GAPDH was used as loading control. Statistical differences are indicated as * for *p* value < 0.05, ** for *p* value < 0.01, *** for *p* value < 0.001 and **** for *p* value < 0.0001.

**Figure 5 biomedicines-10-01835-f005:**
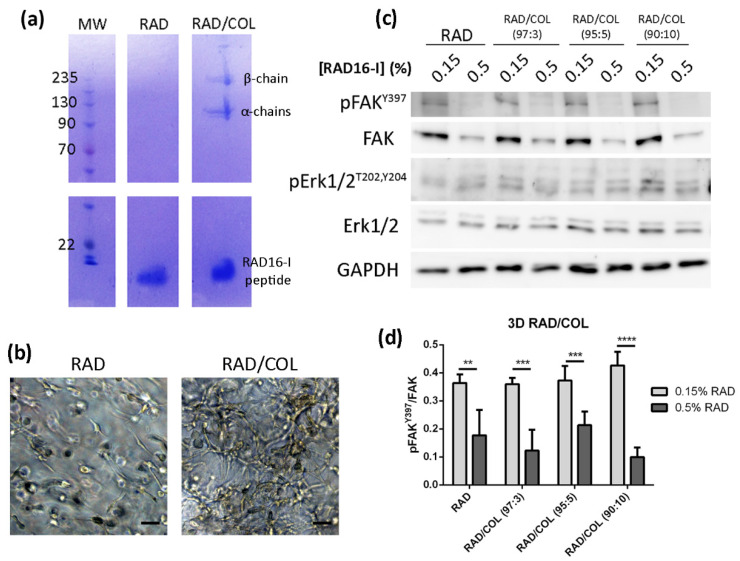
Cell culture in RAD16-I matrix functionalized with collagen type I. (**a**) SDS-PAGE of RAD or RAD/COL cell-free gels; (**b**) phase-contrast microscopy images of fibroblast 24 h after cell embedding within 0.15% RAD or RAD/COL gels. Scale bars represent 50 µm; (**c**) Western blot bands of FAK and Erk1/2 and its phosphorylated forms in PANC-1 cells cultured in 0.15% (soft) and 0.5% (stiff) RAD16-I gels mixed with increasing amounts of collagen I; (**d**) densitometry of pFAK^Y397^/FAK shown in (**c**). GAPDH was used as loading control. Statistical differences are indicated as ** for *p* value < 0.01, *** for *p* value < 0.001 and **** for *p* value < 0.0001.

**Figure 6 biomedicines-10-01835-f006:**
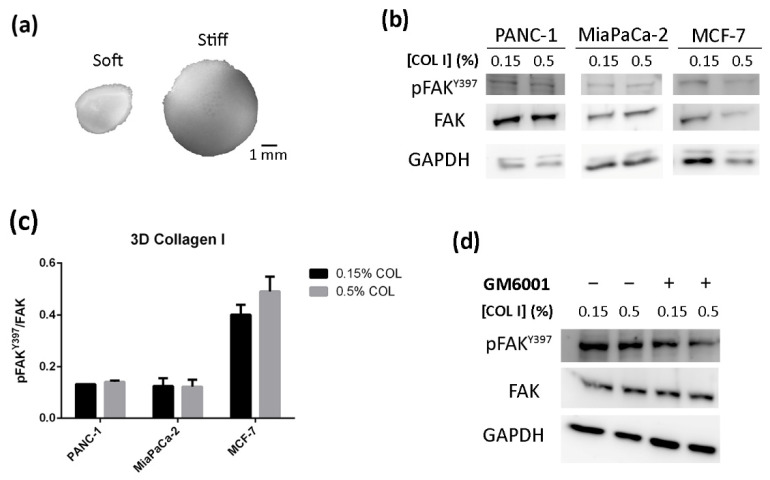
Cell culture in 3D collagen type I gels. (**a**) Macroscopic view of collagen 3D constructs containing PANC-1 cells; (**b**) Western blot bands of pFAK^Y397^ and total FAK in PANC-1, MiaPaCa-2 and MCF-7 cells cultured in 0.15% (soft) and 0.5% (stiff) collagen gels; (**c**) densitometry of bands shown in (**b**) represented as pFAK^Y397^/FAK; (**d**) Western blot bands of pFAK^Y397^ and total FAK in PANC-1 cells cultured in 0.15% (soft) and 0.5% (stiff) collagen gels in the presence of GM6001.

## Data Availability

Data are contained within the article.

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
