# Peer review of "Increased Stiffness Downregulates Focal Adhesion Kinase Expression in Pancreatic Cancer Cells Cultured in 3D Self-Assembling Peptide Scaffolds"

_biomedicines, 2022, doi:10.3390/biomedicines10081835_

Round 1

Reviewer 1 Report

In this manuscript, authors developed a 3D cell culture model using a synthetic peptide-based matrix to study the effect of matrix stiffness on PDAC in terms of FAK expression. They found that increasement of stiffness promoted FAK downregulation of PDAC cells in RAD16-I scaffold. Each experiment was well designed and the experimental method was clearly described. However, it is difficult to understand exactly what the significance of this study is. In my opinion, the synthetic self-assembled peptide nanofiber matrix, RAD16-I, seems to be the key point of this paper. The authors need to clearly demonstrate the advantages of 3D-culture with RAD16-I over other previous studies.

The abstract needs to be more concise and clearly reveal the purpose of this study, and it is necessary to clearly describe the new aspects of this study's progress from the previous reports.

In addition, it would be better to discuss the physiological relevance of the synthetic self-assembled peptide nanofiber matrix 3D culture model.

Reviewer 2 Report

Summary

This paper entitled “Increased stiffness downregulates focal adhesion kinase ex-2 pression in pancreatic cancer cells cultured in 3D self-assembling peptide scaffolds” by Betriu et al. investigated the effect of matrix stiffness in pancreatic ductal adenocarcinoma cells by performed a series in vitro experiment on RAD16-I gels and studied the possible mechanism of the focal adhesion kinase.

Their main finding including:

-        FAK activation is independent with matrix stiffness in RAD16-I

-        Increased matrix stiffness reduced FAK expression

-        FAK expression remains the same regardless of stiffness with collagen

Many studies have confirmed that stiffer PDAC cells are more invasive. However, the mechanism is not clear especially in molecular level. This study has studied the molecular mechanisms on an artificial platform. This is an interesting study though some of the result is expected.

The overall quality of this work is very good. The experiment design is good, and proper reference is conduct. There is fair amount of experiment has been performed. The manuscript is well-prepared. Data is convincing and well discussed. Although the experiment design good and data is convincing, there are some of the concerns will be addressed as follows:

Major concerns

1 in Figure 4, pFAKY397, FAK and the ratio have been studied. However, only ratio has been studied in the Fig 5 and 6, would you mind explain why the experiment been designed?

2 Would you mind explain why the test cell lines are switching between hNCF and MCF-7?

3 Why DMA analysis stops at day 10? Is 10 days enough to test the changes? What is the cell concentration in the gel?

4 In figure 4, why hNDF disappeared in the following analysis?

Minor Issues

1 Although authors have mentioned statistical differences sign in statistics section, it will be more convenience for readers if further clarify in figure caption.

2 Figure 6a is not clear enough to support authors’ claim that cell is dragging the hydrogel. Further explanation will be appreciated.
